# Characterization of Individuals Interested in Participating in a Phase I SARS-CoV-2 Vaccine Trial

**DOI:** 10.3390/vaccines9101208

**Published:** 2021-10-19

**Authors:** Parichehr Shamsrizi, Frederik Johannes Kramer, Marylyn Martina Addo, Anahita Fathi

**Affiliations:** 1Department of Medicine, Division of Infectious Diseases, University Medical Center Hamburg-Eppendorf, 20246 Hamburg, Germany; p.shamsrizi@uke.de (P.S.); 61807207@edu.kl.ac.at (F.J.K.); m.addo@uke.de (M.M.A.); 2Department for Clinical Immunology of Infectious Diseases, Bernhard Nocht Institute for Tropical Medicine, 20359 Hamburg, Germany; 3German Center for Infection Research, Hamburg-Lubeck-Borstel-Riems, 38124 Hamburg, Germany; 4Medical School, Karl Landsteiner University of Health Sciences, 3500 Krems an der Donau, Austria

**Keywords:** SARS-CoV-2, COVID-19, vaccine, clinical trial, recruitment, motivation, demographic, MVA-SARS-2-S, phase 1

## Abstract

The development of an effective vaccine against SARS-CoV-2 marks one of the highest priorities during the ongoing pandemic. However, recruitment of participants for clinical trials can be challenging, and recruitment failure is among the most common reasons for discontinuation in clinical trials. From 20 May 2020, public announcements about a planned phase I trial of the vaccine candidate MVA-SARS-2-S against SARS-CoV-2 began, and interested individuals started contacting the study team via designated e-mail. All emails received from these individuals between 20 May 2020–30 September 2020 were assessed. Of the 2541 interested volunteers, 62% contacted the study team within three days after the first media announcement. The average age was 61 years (range 16–100), 48% of volunteers were female and 52% male. A total of 274, 186, and 53 individuals, respectively, reported medical pre-conditions, were health-care workers, or had frequent inter-person contacts. In conclusion, we report a high number of volunteers, with a considerable percentage stating factors for an elevated risk to acquire COVID-19 or develop severe disease. Factors such as media coverage and the perception of a disease as an acute threat may influence the individual’s choice to volunteer for a vaccine trial. Our data provide first important insights to better understand reasons to participate in such trials to facilitate trial implementation and recruitment.

## 1. Introduction

The development of an effective vaccine against SARS-CoV-2, the causative agent of Coronavirus Disease 2019 (COVID-19), marks one of the highest priorities during the ongoing pandemic. However, recruitment of participants for clinical phase I vaccine trials can be challenging, and recruitment failure is among the most common reasons for discontinuation in clinical trials [1]. The media plays an important role in informing the public and may raise awareness about upcoming vaccine trials [2]. Furthermore, the understanding of individual motivations and demographic characteristics of potential trial participants may facilitate recruitment and the implementation of clinical trials.

The University Medical Center Hamburg-Eppendorf and CTC North, a tertiary care hospital and clinical research organization, respectively, conducted a phase 1 trial of the vector vaccine candidate MVA-SARS-2-S against COVID-19 between October 2020 and July 2021 (registered at ClinicalTrials.gov NCT04569383). The trial was designed as an open-label trial to assess the safety, tolerability, and immunogenicity of two ascending doses of the candidate vaccine in 30 healthy men and women between 18–55 years. Participants received two vaccinations of either a lower or a higher vaccine dose (*n* = 15, respectively) on days 0 and 28 and adverse events, laboratory values, and immune responses were assessed during multiple ambulatory visits throughout the study duration of six months.

From 20 May 2020, various German media outlets began reporting about the trial and encouraged interested individuals to contact the study team via email. These first reports included general information, i.e., that healthy adults 18 aged years or older were planned to be included and an overview about the rationale of the study and the number of participants needed. Detailed information about financial compensation, the study duration and the number of visits were published in late September 2020, after the approval of the study by the responsible ethics committee. In this report, we focused on the responses to the early media reports. We assessed the volunteers’ age, sex, and health status, and infer possible reasons for motivation, to gain an understanding of the characteristics of individuals interested in the participation of this early clinical vaccine trial in the context of the COVID-19 pandemic.

## 2. Materials and Methods

All e-mails received between 20 May 2020 and 30 September 2020 by the study team’s email account were assessed. The respective email address had been published by various online, print, and TV media. The content of these e-mails was unsolicited. Whenever an individual expressed their interest to participate in the planned phase I trial, available information about age, sex, pre-existing medical conditions and additional information (e.g., reasons for interest in participation) was extracted and then analyzed anonymously and in a descriptive fashion. Based on the definition of risk age for a severe course of COVID-19 by the World Health Organization (WHO) [3], individuals who stated their age were further divided into a group with elevated risk (≥60 years old) and a group without elevated risk (<60 years old).

In addition, concerning medical preconditions, individuals were separated into the following subgroups:No medical preconditions;Medical preconditions not associated with an increased risk of a severe course of COVID-19;Medical preconditions associated with an increased risk of a severe course of COVID-19;No information about medical preconditions provided.

The definition of medical preconditions associated with an increased risk of a severe course of COVID-19 was based on the definition published by the Robert Koch Institute (the German national public health agency) and included cardiovascular, pulmonary, rheumatological and autoimmune disease, diabetes mellitus, malignancies, obesity and immunosuppression [4]. Whenever further information, e.g., reasons for motivation to participate was stated, this information was collected and is likewise reported here. A comparison between further information provided in e-mails of the first days and in e-mails of the remaining time was assessed in addition.

Furthermore, we illustrated the COVID-19 situation in Hamburg, Germany during the time the data were collected. We used the daily case numbers released by the Robert Koch Institute [5].

Data were extracted with Microsoft Excel (version 16.53^®^, Microsoft, Redmond, Washington, DC, USA). Descriptive analysis was performed with STATA (version 16.1^®^, StataCorp LLC, College Station, TX, USA). Microsoft PowerPoint (version16.53^®^, Microsoft, Redmond, Washington, DC, USA), was used for data visualization.

## 3. Results

The observational period of the study was from 20 May 2020 to 30 September 2020. In this time period, COVID-19 case numbers in Hamburg Germany were declining, starting from a number of around 300 daily new cases at the end of April 2020 to 7 new daily cases in July 2020. The situation is illustrated in Figure 1.

A total of 2541 individuals interested in participation in the vaccine trial contacted us between 20 May 2020 and 30 September 2020, and their information was extracted and included in this analysis. The majority of the individuals (*n* = 1568; 62%) contacted us within the first three days following the first press release on 20 May 2020: 995 (39%), 316 (12%), and 257 (10%) on days 1–3, respectively.

Of the 2541 volunteers, 1334 (52%) were men and 1207 (48%) were women (Figure 2a).

In total, 1582 (62%) of the volunteers commented on their age, and 1561 (61%) of the volunteers stated their exact age. The range was from 16 to 100 years, with an average age of 61 years (Figure 2b). Two individuals indicated that they were over 60 years old. In total, 501 individuals were over 60 years old and 1062 were under 60 years old. Nineteen individuals only indicated that they were over 18 years old and 959 individuals did not mention their age (Figure 2c).

Concerning health status, 918 (36%) provided information, whereas 1623 (64%) did not. Further, 274 (11%) stated underlying medical preexisting conditions; of these, 203 (74%) individuals reported preconditions that constitute a high risk for a severe course of COVID-19. Finally, 644 (23%) of all individuals explicitly stated that they were healthy (Figure 2d).

Further information about their background and reasons for motivation was provided by 304 individuals, whereas 2237 individuals did not provide any further information.

Of the volunteers, 186 individuals stated that they were health care workers (HCW). A high number of inter-person contacts was mentioned by 53 individuals who contacted us (e.g., due to their professions, such as teachers or police officers). In addition, 30 individuals indicated that they regularly participated in clinical trials. Financial reasons were mentioned in 21 cases; altruistic reasons in 8 cases (i.e., helping society, protecting relatives, or supporting science). One individual was a journalist and motivated to report about the experience of being a participant, and another stated that he was scientifically interested. Three individuals mentioned more than one reason (Figure 2e).

A comparison between the indicated reasons provided by participants who contacted us within the first three days and the reasons provided by the ones who contacted us in the remaining time period shows a similar distribution (Figure 2g).

## 4. Discussion

Recruitment of participants for vaccine trials is usually challenging [6]. The strong interest of individuals to participate in a clinical phase I SARS-CoV-2 vaccine trial we described here is encouraging. This may have several reasons. The COVID-19 pandemic and corresponding vaccine trials have received significant media coverage, which may have mobilized and motivated the public to participate in such trials [7]. We received the majority of inquiries within three days of first media coverage of the trial. In a Germany-wide survey of around 1000 participants in December 2020, 64% mentioned television and radio as one of their preferred source for information about vaccines against SARS-CoV-2 [2]. Even though the local case numbers were decreasing at the beginning of our observed time period, the pandemic represents an exceptional and unprecedented situation which is perceived as an acute threat by a significant proportion of the population, specifically persons at risk for a severe disease course [8,9]. We noted that a considerable number of interested individuals belonged to a group with a disproportionate risk either for a severe disease course (i.e., older age or specific pre-conditions) or for acquisition/transmission of disease (i.e., health care workers or individuals with frequent social interactions). This distribution does not differ much when comparing the data from the first days with the data from the rest of the time (Figure 2f,g). With regard to COVID-19 vaccine trials specifically, Germany-wide surveys showed that around 20–25% of respondents would participate in a SARS-CoV-2 vaccine trial (personal communication C. Betsch/P. Giesler, manuscript in preparation).

To date, only few studies have systematically assessed the characteristics and reasons that motivate individuals to participate in clinical trials. While patients participating in clinical drug trials may experience a direct medical benefit from participating, this direct benefit is usually not the main incentive for participation in preventive vaccine trials [6]. A 2021 review on characteristics of vaccine trial participants in the US found that the sex distribution was 44.8% male vs. 55.2% female across 40 trials [10]. We observed a similarly balanced trend. In a recent review by Detoc et al. assessing age in the context of motivation to participate in US-based vaccine trials, the mean age of volunteers was found to be 38.1 years, and individuals who were not interested in participation were significantly older (54.9 years) [11]. The elderly are often underrepresented in clinical research, not only due to exclusion criteria, but also for more complex reasons such as social and cultural barriers or physical impairments [12]. It is, therefore, encouraging to see a strong interest of older individuals to participate in the MVA-SARS-2-S trial. Regarding motivation to participate in clinical trials, Detoc et al. found that monetary compensation was a key incentive for up to 67% of healthy individuals [11] and this observation has been made by others [13]. Interestingly, only 21 individuals mentioned financial reasons for participation in the correspondence we received.

This retrospective observational study has notable limitations. The emails we received were unsolicited, therefore, a systematic assessment of the individuals’ characteristics such as health status and reasons for interest in trial participation is not possible. While the information the individuals provided suggests possible motivational reasons, these cannot be confirmed, and it may well be that an elderly person who has an elevated risk for severe COVID-19 may be interested in a trial participation for financial or other reasons (and not due to their risk profile). Still, the information we gathered here provides important first insights. We believe that further studies are urgently needed to better understand motivational factors of potential trial participants and ultimately promote participation in early phase vaccine trials as a means to confront emerging infectious disease threats.

## Figures and Tables

**Figure 1 vaccines-09-01208-f001:**
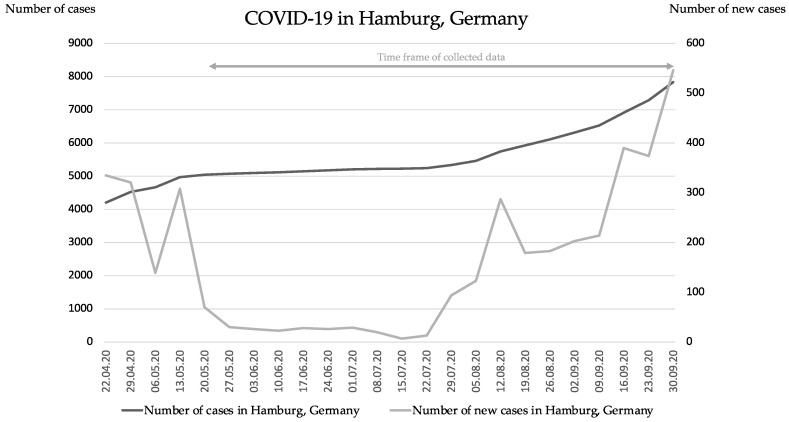
COVID-19 numbers in Hamburg, Germany: Weekly number of cases and weekly number of new cases.

**Figure 2 vaccines-09-01208-f002:**
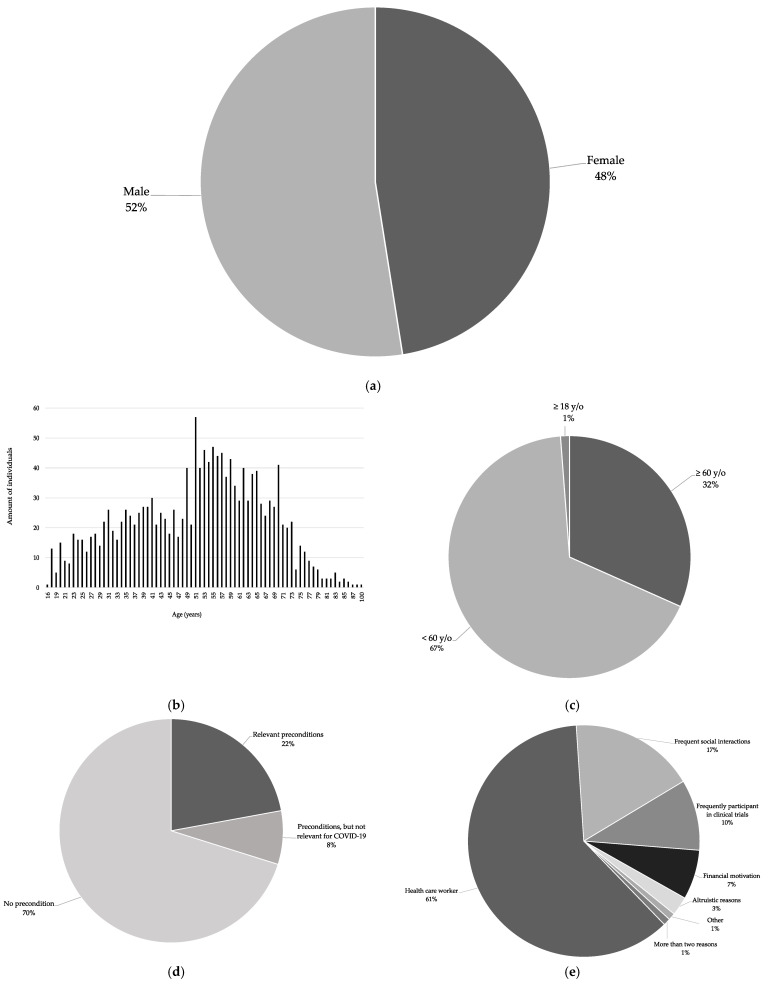
Unsolicited information provided by volunteers. “*n*” denotes the number of individuals providing information. (**a**) Sex distribution (stated by *n* = 2541/2541); (**b**) Age distribution (age stated by *n* = 1561/2541); (**c**) Age distribution divided as under 60 over 60 years old individuals (age represents a separate risk factor for severe disease of COVID-19 (stated by *n* = 1582/2541); (**d**) Preexisting medical conditions (stated by *n* = 918/2541); (**e**) Specific information of volunteers implying potential reasons for interest (stated by *n* = 304/2541); (**f**,**g**) Specific information of volunteers implying potential reasons for interest divided into: First three days (*n* = 218/1661) vs. the remaining period (*n* = 86/880).

## Data Availability

The data presented in this study are available on request from the corresponding author.

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
