# Peer review of "Characterization of Individuals Interested in Participating in a Phase I SARS-CoV-2 Vaccine Trial"

_vaccines, 2021, doi:10.3390/vaccines9101208_

Round 1

Reviewer 1 Report

The manuscript entitled ‘characterization of individuals interested in participating in a phase I SARS-CoV-2 vaccine trial’ by Shamsrizi et al. in interested. In this manuscript, the author analyzed the interest of individuals to participate in PHASE  1 clinical trials. The author carefully documented the data and an excellent analysis of the results is presented. Hence, this manuscript is suitable for publication. However, the author needs to address the following minor comments before to publish

Comment to author

  1. Authors need to discuss the scope of high payment which may encourage some of the individual, the author need to discuss this aspect
  2. As Covid-19 majorly effective against persons with age above 50, but there are enough number of individual above 60 also participated, what is the importance of including them in study or why they are interested to participate even there is a chance of high risk
  3. Figure 1 can be split into two individual figures and data can be presented more impressive way or designing the attractive graphics to attract the readers
  4. What about the feedback of the participant after the end of the study need to consider whether they feel good or not?
  5. Abstract need to be more clear about the importance of this manuscript
  6. Conclusions need to be improved
  7. Authors need to cite the following literature

https://doi.org/10.1016/j.addr.2021.01.002

https://doi.org/10.1007/s11481-020-09981-0

  1. What was the major motivation for participants who response within 3 days and who response in the last few days?
  2. How the media announcement helps to participant
  3. What can be an alternative strategy to improve the number of participants needed if any for future diseases/pandemics
  4. Is any participant who discontinue in between the study and what could be the reason

Reviewer 2 Report

Dear Authors,

Our congratulations to the authors for the efforts made in scientific research.

After multiple readings of your manuscript, we think this is a well-conducted study that describes the characteristics of the participants enrolled in a phase I Sars-Cov-2 vaccine trial.

We make some suggestions just to enhance readability.

  1. Both in the abstract and in the second paragraph of the introduction (we could not find the line numbers on the side of the page, so pardon if we are vague) you say “from May 20th” without specifying the year. Even though the reader can deduct this is referring to 2020, we believe it would be better to report a complete date.
  2. In the second paragraph of the introduction, it is reported “are currently conducting”. A reader may be misled by the fact that publication date could be quite far from the writing date of the article. We suggest to reformulate this sentence avoiding relative time references.
  3. The reference [2] is not clear. Moreover, it does not provide a real reference to verify the contents. Please provide a journal, an URL or some other way to access the content.
  4. Figure 1: we think in overall this is not much informative. Firstly, in our version of the manuscript the pie charts are not colored, as well as the legends, making impossible understanding the contents without consulting the text. Second, the (b) figure describe the distribution of age. The reader, however, when reading the text, may expect a pie chart of age like the ones provided for the other characteristics. This happens because the authors divided the respondents into two groups (elevated risk/ not elevated risk of severe COVID course) according to age (with 60 years of age as cutoff value) but they do not provide a graphical representation of this finding, even though they correctly discuss it. Lastly, we think the raw number are necessary but not sufficient for a pie chart. We suggest integrating the percentage to obtain a first-sight readability.

Kind Regards,

Reviewer 3 Report

The manuscript is well-written with no noticeable flaws, but please clarify the following points.

  1. Any information that might have affected the decision of the responders should be presented. A) What kind of information on the trial, especially on the eligibility, financial rewards, or expected efficacy, did the responders have when they responded? Because this information would strongly affect their decision on whether or not to respond, it should be presented as much as possible. Did the investigators announce the recruitment on their web site? If so, what kinds of information were there? If media were the only source, what did the media report to the citizens?  B) During the study period, how serious was the epidemic of COVID-19 in Hamburg, Germany?  C) What was the availability of other vaccine trials? If any competing trials were present, was there any difference that could have affected the choice of the respondents?
  2. There is little information on the trial except for being a phase I SASR-CoV-2 trial. Brief summary of the protocol should be presented with the link to the registration site if any.
  3. In Figure 1, it is unclear which part of the pie charts legend corresponds to which item. Please color or shade them to make the correspondence distinguishable.

Author Response

This manuscript is a resubmission of an earlier submission. The following is a list of the peer review reports and author responses from that submission.